# Boosting Catalytic Combustion of Ethanol by Tuning Morphologies and Exposed Crystal Facets of α-Mn₂O₃

Wangwang Liu [1], Yong Men [1,2,*], Fei Ji [1], Feng Shi [1], Jinguo Wang [1], Shuang Liu [1], Tamerlan T. Magkoev [3,*] and Wei An [1,*]

[1] School of Chemistry and Chemical Engineering, Shanghai University of Engineering Science, Shanghai 201620, China
[2] Mechanical Industrial Key Laboratory of Boiler Low-Carbon Technology, Shanghai University of Engineering Science, Shanghai 201620, China
[3] Laboratory of Surface Physics and Catalysis, Department of Condensed Matter Physics, North Ossetian State University, Vatutina 44-46, Vladikavkaz 362025, Russia
* Correspondence: men@sues.edu.cn (Y.M.); t_magkoev@mail.ru (T.T.M.); weian@sues.edu.cn (W.A.); Tel.: +86-21-6787-4046 (Y.M. & W.A.); Fax: +86-21-6779-1214 (Y.M. & W.A.)

**Abstract:** Three types of α-Mn₂O₃ catalysts with different well-defined morphologies (cubic, truncated octahedra and octahedra) and exposed crystal facets have been successfully prepared via hydrothermal processes, and evaluated for ethanol total oxidation with low ethanol concentration at low temperatures. The α-Mn₂O₃-cubic catalyst shows a superior catalytic reaction rate than that of α-Mn₂O₃-truncated octahedra and α-Mn₂O₃-octahedra under high space velocity of 192,000 mL/(g·h). Based on the characterization results obtained from XRD, BET, FE-SEM, HR-TEM, FT-IR, H₂-TPR, XPS, ethanol-TPD, and CO-TPSR techniques, the observed morphology-dependent reactivity of α-Mn₂O₃ catalysts can be correlated to the good low-temperature reducibility, abundant surface $Mn^{4+}$ and adsorbed reactive oxygen species, which was originated from the exposed (001) crystal planes. Through tuning the morphology and exposed (001) crystal facet of α-Mn₂O₃, a highly active ethanol oxidation catalyst with high selectivity and excellent stability is obtained. The developed approach may be applied broadly to the development of the design principles for high-performance low-cost and environmentally friendly Mn-based oxidation catalysts.

**Keywords:** ethanol combustion; manganese oxide; morphology effect; structure-sensitive

## 1. Introduction

As atmospheric and photochemical contaminants, Volatile Organic Compounds (VOCs) emitted from fossil fuel combustion, transportation, automobile exhaust, and photochemical pollution have caused tremendous detrimental impact on the environment and human's health [1,2]. As one of the typical gaseous pollutants among these VOCs, ethanol derived mainly from the residue of unburned ethanol in ethanol-fueled vehicles [3,4]. By comparing different technologies for burning ethanol into $CO_2$, $H_2O$, and other less hazardous by-products, the catalytic ethanol combustion has been recognized as one of the most viable and environmentally-friendly technologies at low concentration and lower temperature (200–600 °C) owing to its higher efficiency and yields, and lower cost than that of traditional physical/chemical adsorption and non-catalytic thermal oxidation technologies [5]. Thus, the development of the efficient ethanol combustion catalysts is of great significance in reducing ethanol emissions below the limits imposed by the regulation standards.

In the past few decades, noble metal-based catalysts such as Pt, Pd, Ir, Rh and Au [6–10], etc., have been reported to exhibit high performance of ethanol oxidation at low operation temperatures. However, the high cost of noble metals have caused bottlenecked, which limits the economic viability and impedes the widespread use in ethanol-fueled vehicles. Therefore, it is highly desirable to develop an alternative catalyst to substitute for

the noble-metal-based catalyst in order to meet the ever-rigorous ethanol emission standards. Metal oxides, such as $Cu/Al_2O_3$ [11], $CuO/Fe_2O_3$ [3], Al/Mn-K [12], $CoFe_2O_4$ [13], Mn-Ce-Zr-O [14], Mn-Cu [15], etc., are considered as the most promising alternative materials, which have been validated via their good catalytic activity, low cost, resistant to poisonings and higher thermal stability. Catalyst composition, specific surface area, crystal and pore structure have been reported to be the key factors in determining the ethanol combustion efficiencies. Despite its excellent catalytic performance, most of the studies were performed to work the catalysts under lower space velocity, remaining insufficient for practical application.

During recent years, more and more researchers have paid their attention to nano/microsized metal oxides material with different morphologies, such as tube, rods, wires, spheres, cubic, and octahedra, etc. Generally, the catalytic reaction performance of catalyst particles can be finely tuned by their anisotropic morphology, which further results in different exposed crystal facets, to this end, the degree of coordination unsaturation of catalytically active atoms are vital for the correlation between structure and catalytic performance, as reported in the earlier literatures [16–22]. For instance, Shen et al. [21] investigated that the catalytic oxidation of CO over nanoscale catalytic particles of $Co_3O_4$ with controlled size and topology, and found that $Co_3O_4$ nanorod have highly structure-ordered, constituting 40% of the (110) crystal plane on the nanorod surface was able to oxidize CO at $-77\,^\circ$C, showing a 10 times higher activity than conventional $Co_3O_4$ catalysts. Trovarelli et al. [19] reported that soot combustion over ceria was a surface-dependent reaction and (100) surface for nanocubes; (100), (110) and partially (111) for nanorods, and enclosed octahedral possessed much higher catalytic performance for soot oxidation than ceria polycrystalline powders, which in agreement with the density functional theory (DFT) calculations that the formation energies of ceria surface oxygen vacancies decreased tendency were obtained: (111) < (100) < (110) [23]. Hence, morphology-controlled synthesis of nano/microsized metal oxides further brings up new opportunities for tuning the catalytic activity, selectivity, and stability of metal oxide catalysts via selectively exposing uniform and higher energy/reactive crystal facets.

For transition-metal oxides (TMOs), the Mars-Van Krevelen (MVK) mechanism is proverbially accepted to be responsible for the oxidation of VOCs; the gas-phase organic molecules are oxidized by the active oxygen species of TMOs, then re-oxidized by gas-phase oxygen molecules, which will regenerate or maintain the oxidation state of the metal cations [24,25]. The MVK mechanism is mainly caused by transition-metal cations which possess the ability of electron transport or lattice oxygen mobility for their d or f outer electrons. As an important TMOs, earth-abundant manganese oxides ($MnO_x$), such as $MnO_2$, $Mn_2O_3$, and $Mn_3O_4$, are considered as promising catalytic materials due to their potentially high catalytic performance, low cost, low toxicity, and durable for the catalytic oxidation of VOCs [1,26–31], soot [32,33], or CO [34–36]. The superior efficiency catalytic reactivity of $MnO_x$ material mainly related to the existence of various valence states of manganese ions ($Mn^{2+}$, $Mn^{3+}$, and $Mn^{4+}$ species) and lattice oxygen on the $MnO_x$ materials, resulting in a facile and reversible $Mn^{3+}/Mn^{2+}$ or $Mn^{4+}/Mn^{3+}$ redox cycle [25,32,37–39]. For example, Gandhe et al. [39] revealed that the total oxidation performance of ethyl acetatethe over cryptomelane type octahedral molecular sieve (OMS-2) material was dependent on the existence of $Mn^{4+}/Mn^{3+}$ type redox couples and facile lattice oxygen on catalysts. Peluso et al. [40] reported that high concentration of $Mn^{3+}$ was beneficial to weakening Mn-O bond and increasing the concentration of active oxygen species, which would enhance catalytic performance of $MnO_x$ in the catalytic oxidation of ethanol. Kim et al. [37] reported that VOCs oxidation over manganese oxides catalysts, including $Mn_3O_4$, $Mn_2O_3$ and $MnO_2$ follows $Mn_3O_4 > Mn_2O_3 > MnO_2$ could be correlated to the surface area and oxygen mobility of samples. Hence, the multiple valence states and surface reactive oxygen species of $MnO_x$ catalysts are the major determining factors for the catalytic activity of VOCs oxidation.

So far, much research efforts have been devoted to further develop high-performance $MnO_x$ catalysts with excellent reactivity and selectivity by tailoring the shape of the catalysts [20,32,41,42]. Feng et al. [43] reported different manganese oxides with distinct morphologies (1D-$Mn_3O_4$ nanorod, 2D-$Mn_3O_4$ nanoplate, and 3D-$Mn_3O_4$ nano-octahedron) were synthesized by hydrothermal treatment. The $Mn_3O_4$ nanoplate catalyst exhibits a small crystal size, large surface area, more exposed (112) facets, abundant $Mn^{4+}$, and defective structure, contributing to a superior catalytic performance at the high space velocity of 120,000 mL $g^{-1}$ $h^{-1}$. Wang et al. [41] reported shape-dependent activation of peroxy monosulfate by single crystal $\alpha$-$Mn_2O_3$ (cube, octahedra and truncated octahedra) for catalytic phenol degradation in aqueous solution followed the order of catalytic activity of three $\alpha$-$Mn_2O_3$ samples as $\alpha$-$Mn_2O_3$-cubic > $\alpha$-$Mn_2O_3$-octahedra > $\alpha$-$Mn_2O_3$-truncated. Recently, [32] crystal facet-dependent reactivity of $\alpha$-$Mn_2O_3$ microcrystalline catalyst for soot combustion was reported, in the rank of $\alpha$-$Mn_2O_3$-cubic > $\alpha$-$Mn_2O_3$-truncated octahedra > $\alpha$-$Mn_2O_3$-octahedra. The origin of the superior performance of cubic $\alpha$-$Mn_2O_3$ was correlated with the higher concentration of low-coordinated surface oxygen sites and improved surface redox properties on deliberately exposed (001) crystal facets.

To the best of our knowledge, the ethanol total combustion over different morphologies $\alpha$-$Mn_2O_3$ materials with selectively exposed different crystal facets has not yet been investigated in the literature. In the presented work, three types of controllably synthesized three types of $\alpha$-$Mn_2O_3$ catalysts with different morphology and exposed crystal facets were prepared by a facile hydrothermal route. The intrinsic properties of $\alpha$-$Mn_2O_3$ catalysts are characterized by means of XRD, BET, FE-SEM, HR-TEM, FT-IR, $H_2$-TPR, XPS, Ethanol-TPD, and CO-TPSR techniques. Kinetic study was also performed to understand the morphology-dependent reactivity of $\alpha$-$Mn_2O_3$ catalysts for ethanol total combustion. As demonstrated by various physicochemical characterizations, the morphology of $\alpha$-$Mn_2O_3$ catalysts was identified to play a crucial role in exposing the different crystal facets and therefore dictate the catalytic performance of the ethanol total combustion.

## 2. Results and Discussion

### 2.1. Crystal Phase Structure and Morphology of Catalysts

The crystal phase of as-prepared manganese oxides catalysts was confirmed by XRD, whose patterns were presented in Figure 1. All catalysts showed the same main characteristic diffraction peaks with a remarkable crystallinity, agreeing well with the previous work [32,41,44,45]. The main characteristic diffraction peaks at 23.1°, 33.0°, 38.2°, 45.2°, 49.3°, 55.2°, and 65.9° (2θ values), corresponding to the (211), (222), (400), (332), (431), (440), and (622) (hkl) planes, can be well-indexed to the body centered cubic phase crystalline structure of $\alpha$-$Mn_2O_3$ with the lattice parameter a = 0.9409 nm (JCPDS Card No:00-041-1442). From the diffraction profiles of $\alpha$-$Mn_2O_3$-C, $\alpha$-$Mn_2O_3$-TO, and $\alpha$-$Mn_2O_3$-O, one can observe the considerably sharpening of the characteristic diffraction peaks with the increasing relative intensity, indicating the enhanced crystallinity. The crystalline sizes of $\alpha$-$Mn_2O_3$-C, $\alpha$-$Mn_2O_3$-TO and $\alpha$-$Mn_2O_3$-O were estimated to be 24.8 nm, 51.6 nm and 76.1 nm, respectively. No diffraction peaks of other impurities phase were observed via the XRD patterns, further implying that three catalysts possessed high phase purity.

The morphologies and crystallographic microstructures of $\alpha$-$Mn_2O_3$ catalysts were investigated by using FE-SEM and HR-TEM. Figure 2 clearly showed that three catalysts possessed the same $Mn_2O_3$ crystal phase but presented completely different morphologies, including cubic, truncated octahedra, and octahedra, implying the morphologies can be controlled by adjusting hydrothermal temperatures and different solvents. The observations were quite similar to the previous works [32,41]. The $\alpha$-$Mn_2O_3$-C presented well-regulated cubic morphology with amiable edges and particle sizes ranging between 0.7 and 2.0 μm (Figure 2a,b). Based on these FE-SEM images (Figure 2c–f), the $\alpha$-$Mn_2O_3$-TO and $\alpha$-$Mn_2O_3$-O catalysts clearly presented uniform distribution of truncated octahedra and octahedra morphologies with the smooth surface and sharp edges and particle sizes ranging from 1.0–2.2 μm and 0.8–2.5 μm, respectively. No particle with other morpholo-

gies was observed in the three samples with cubic, truncated octahedra, and octahedra morphologies (Figure 2a,c,e).

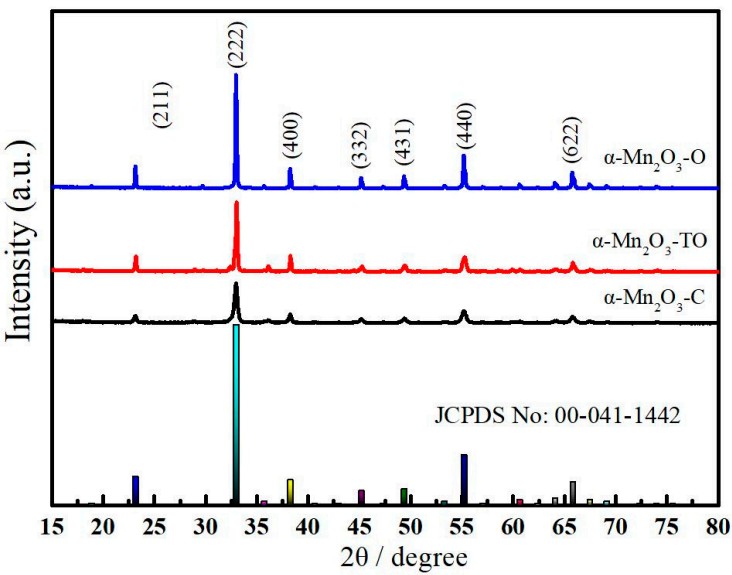

**Figure 1.** XRD patterns of $\alpha$-Mn$_2$O$_3$ catalysts with different morphologies.

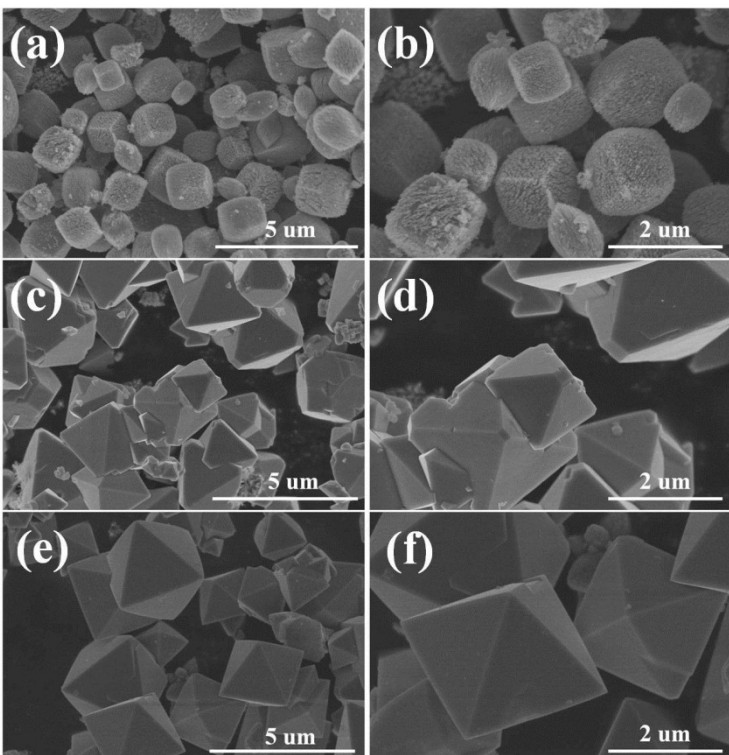

**Figure 2.** FE-SEM images of $\alpha$-Mn$_2$O$_3$ catalysts with different morphologies. $\alpha$-Mn$_2$O$_3$-C (**a,b**), $\alpha$-Mn$_2$O$_3$-TO (**c,d**), $\alpha$-Mn$_2$O$_3$-O (**e,f**).

TEM characterizations were performed on various $\alpha$-Mn$_2$O$_3$ catalysts to further investigate the morphologies and crystallographic features i.e., the exposed crystal facets. TEM and HR-TEM images (Figure 3) clearly showed that three $\alpha$-Mn$_2$O$_3$ catalysts presented morphologies of cubic (Figure 3a), truncated octahedra (Figure 3c), and octahedra (Figure 3e), in consistent with the results of FE-SEM and previous reports [32,41]. Figure 3c showed that the lattice fringe of cubic $\alpha$-Mn$_2$O$_3$ (004) facets was 0.23 nm, and $\alpha$-Mn$_2$O$_3$-C mainly exposed the (001) crystal facets as the crystal (001) facets and (004) facets were parallel

to each other. Figure 3f showed that $\alpha$-$Mn_2O_3$-O mainly exposed the crystal (111) facets which stays parallel to the (222) facets. In addition, the truncated octahedra (Figure 3d) exposed truncated crystal (001) facets and crystal (111) facets, in good agreement with the previous works [32,41,44]. According to the literatures [46,47], crystal growth rates in the direction perpendicular to a high-index plane are usually much faster than those along the normal direction of a low-index plane, therefore high-index planes are rapidly eliminated during particle formation. Li et al. [44] studied the influence of preparation conditions on the morphology control of $\alpha$-$Mn_2O_3$ and found out that the domain relied on the different growth rates of [001] and [111] crystallographic directions. Therefore, growing orientation along the [001] and [111] crystallographic directions can form both morphologies of cubic and octahedra with exposed (001) and (111) crystal facets, respectively.

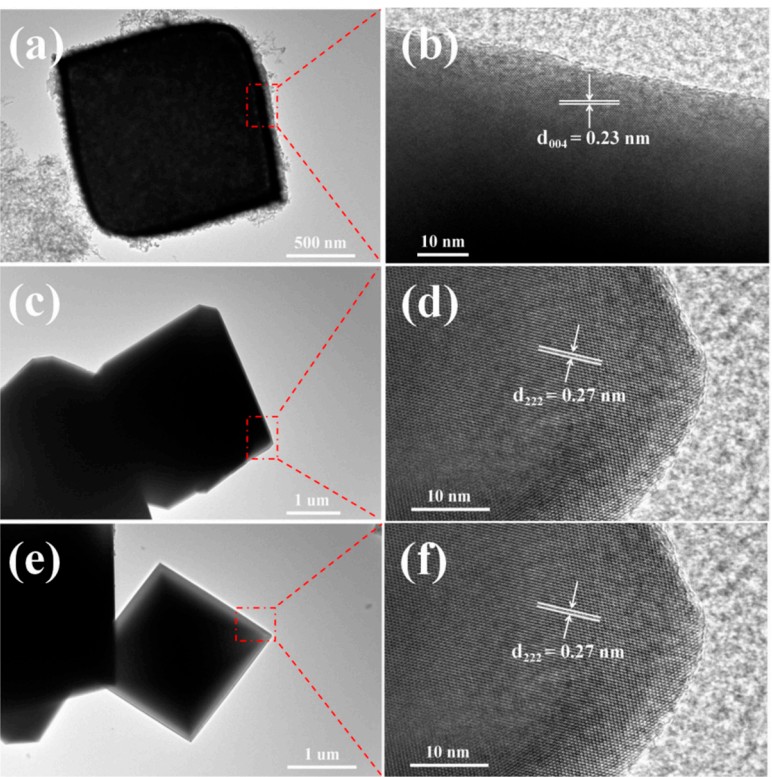

**Figure 3.** HR-TEM images of $\alpha$-$Mn_2O_3$ catalysts with different morphologies. $\alpha$-$Mn_2O_3$-C (**a,b**), $\alpha$-$Mn_2O_3$-TO (**c,d**), $\alpha$-$Mn_2O_3$-O (**e,f**).

FT-IR spectra of the $\alpha$-$Mn_2O_3$ catalysts with different morphologies were shown in Figure S1. It can be seen that the different catalysts presented the analogical positions of characteristic peaks as in literature [32], which located around 481, 529, 572, 660, 1626, 3432 $cm^{-1}$, respectively. In the fingerprint region, the presence of peaks at around 529 $cm^{-1}$, 572 $cm^{-1}$, and 660 $cm^{-1}$ were assigned to the Mn-O-Mn band asymmetric and symmetric stretching vibration and the peak corresponded to metal-oxygen chains (Mn-O band) bending vibrations in $\alpha$-$Mn_2O_3$ [48]. In the functional group region, the weak broad band at around 3432 $cm^{-1}$ was attributed to the stretching vibration modes of the hydroxyl functional group and the peak at around 1626 $cm^{-1}$ was mainly caused by the adsorbed water molecules on the $\alpha$-$Mn_2O_3$.

## 2.2. Surface Area and Surface Chemical Properties

The $N_2$ physical adsorption-desorption isotherms and BJH pore-size distribution patterns of different $\alpha$-$Mn_2O_3$ catalysts are shown in Figure S2 and their $S_{BET}$, $D_p$, and $V_p$ are listed in Table 1. All catalysts displayed a similar type IV isotherms, with H3 ($P/P_0 = 0.6$–$1.0$) hysteresis rings, indicating the presence of microporous and mesoporous

in the $\alpha$-Mn$_2$O$_3$ materials. Table 1 showed that $\alpha$-Mn$_2$O$_3$-C, $\alpha$-Mn$_2$O$_3$-TO, and $\alpha$-Mn$_2$O$_3$-O catalysts possessed surface area of 30.5, 2.5, and 1.0 m$^2$/g, the pore diameter of 22.1, 13.5, 12.9 nm, pore volume of 0.212, 0.009, and 0.004 cm$^3$/g, respectively. Comparing to $\alpha$-Mn$_2$O$_3$-TO and $\alpha$-Mn$_2$O$_3$-O catalyst, $\alpha$-Mn$_2$O$_3$-C had much higher surface area, pore diameter, and pore volume, which is favorable for the activation and diffusion of the reactants, thereby enhancing the catalytic performance of the catalyst. From the adsorption-condition of N$_2$ molecules on the surface of porous material at 77 K, the surface area and pore volume of $\alpha$-Mn$_2$O$_3$-TO and $\alpha$-Mn$_2$O$_3$-O catalysts primarily relied on interparticle spaces, thus both catalysts were seen as less developed porous $\alpha$-Mn$_2$O$_3$ catalysts. The morphologies and structures of catalysts would have great influence on pore size distributions, which is likely to be correlated with the distribution of the active sites [24,49]. From FE-SEM images, one can also observed that all catalysts were composed of solid single crystals with smooth planes, which brought about much smaller surface area, pore diameter, and pore volume.

**Table 1.** Preparation parameters, BET specific surface areas (S$_{BET}$), pore volumes (V$_P$), Pore diameters (D$_P$), reduction peak temperatures, H$_2$ consumptions and surface element composition of three $\alpha$-Mn$_2$O$_3$ catalysts.

| Catalyst | S$_{BET}$ [a] (m$^2$/g) | V$_P$ (cm$^3$/g) | D$_P$ [b] (nm) | T ($^\circ$C) | | H$_2$ Consumptions (mmol/g) | | | Surface Element Molar Ratio [c] | |
|---|---|---|---|---|---|---|---|---|---|---|
| | | | | Peak1 | Peak2 | Mn$_2$O$_3\to$Mn$_3$O$_4$ | Mn$_3$O$_4\to$MnO | Total | Mn$^{4+}$/Mn$^{3+}$ | O$_{ads}$/O$_{latt}$ |
| $\alpha$-Mn$_2$O$_3$-C | 30.5 | 0.212 | 22.1 | 287 | 383 | 2.5 | 3.7 | 6.2 | 1.27 | 0.53 |
| $\alpha$-Mn$_2$O$_3$-TO | 2.5 | 0.009 | 13.5 | 409 | 458 | 3.6 | 2.5 | 6.1 | 1.15 | 0.43 |
| $\alpha$-Mn$_2$O$_3$-O | 1.0 | 0.004 | 12.9 | 449 | 496 | 3.7 | 2.3 | 6.0 | 1.07 | 0.38 |

[a] Specific surface areas were calculated by the BET method. [b] The data was calculated via the BJH method according to the N$_2$ adsorption-desorption isotherms. [c] The ratios were calculated based on the peak areas processed by the XPS-Peak software.

The redox properties of various $\alpha$-Mn$_2$O$_3$ catalysts were examined by using H$_2$-TPR technique. Figure 4 presented the reduction profiles of the $\alpha$-Mn$_2$O$_3$ catalysts in the temperature range from 100 $^\circ$C to 600 $^\circ$C. Two typical peaks can be observed for all catalysts. The different H$_2$-TPR profiles of three $\alpha$-Mn$_2$O$_3$ samples indicated the different reactivity of reduction of reactive oxygen species in different local environments [50,51]. The $\alpha$-Mn$_2$O$_3$-O catalyst exhibited a large reduction peak centered at about 496 $^\circ$C with a low temperature shoulder peak at 449 $^\circ$C, corresponding to H$_2$ consumptions of 2.3 and 3.7 mmol·g$^{-1}$, respectively (Table 1). Comparing with $\alpha$-Mn$_2$O$_3$-O catalyst with exposed crystal (111) facets, a similar reduction pattern was observed for the $\alpha$-Mn$_2$O$_3$-TO catalyst with exposed (001) & (111) crystal facets, which had the primary reduction peak at 458 $^\circ$C and the low temperature shoulder peak n at 409 $^\circ$C, corresponding to H$_2$ consumptions of 3.6 and 2.5 mmol·g$^{-1}$, respectively (Table 1). The alteration of the reduction behavior might be caused by the presence of crystal (001) facets on truncated octahedra $\alpha$-Mn$_2$O$_3$ sample. In the case of $\alpha$-Mn$_2$O$_3$-C catalyst, two distinct lower temperature reduction peaks at 287 and 383 $^\circ$C were observed, corresponding to H$_2$ consumptions of 2.5 and 3.7 mmol·g$^{-1}$, respectively (Table 1). The two reduction peaks of various $\alpha$-Mn$_2$O$_3$ samples corresponds to a stepwise transformation under H$_2$ atmosphere. The first step peak represented the reduction of Mn$_2$O$_3$ to Mn$_3$O$_4$, whereas the second step peak corresponded to the reduction of Mn$_3$O$_4$ to the final state MnO, similar to the findings reported in the previous literatures [27,32,52–56]. The order of low-temperature reducibility for surface oxygen is $\alpha$-Mn$_2$O$_3$-C with crystal (001) facets > $\alpha$-Mn$_2$O$_3$-TO with crystal (001) & (111) facets > $\alpha$-Mn$_2$O$_3$-O with crystal (111) facets, which is related to the fact that surface O atoms on (001) facet, due to its high flexibility, are energetically more facile for release and participation than those on (111) facet [32], suggesting a more facile surface oxygen reducibility for $\alpha$-Mn$_2$O$_3$-C and morphology-dependence of redox behavior. The excellent reducibility of the $\alpha$-Mn$_2$O$_3$ could be beneficial to the enhanced catalytic activity for the ethanol combustion.

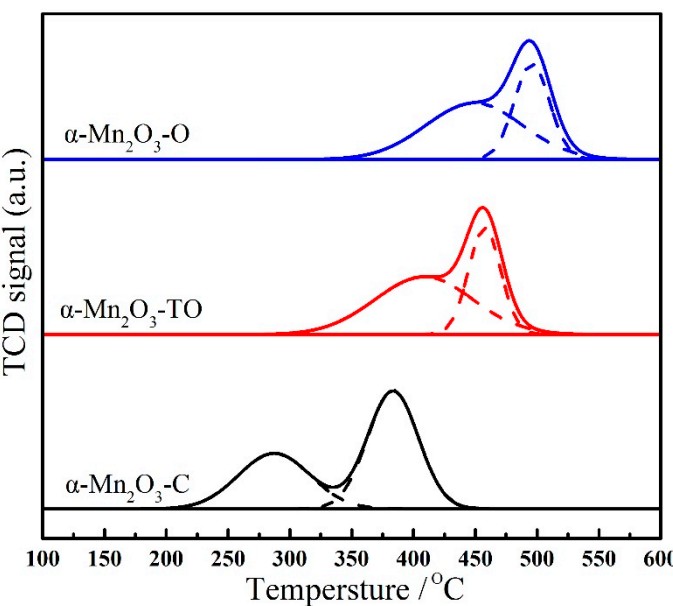

**Figure 4.** $H_2$-TPR profiles of $\alpha$-$Mn_2O_3$ catalysts with different morphologies.

XPS measurements were carried out to identify the surface element compositions, element valence states, and adsorbed oxygen species of $\alpha$-$Mn_2O_3$ catalysts. In the XPS survey spectrum presented in Figure S3A, the peaks of carbon (C 1s), oxygen (O 1s), and manganese (Mn 2p) can be clearly detected. In Figure S3B, the binding energy (BE) obtain at 284.6 eV and 287.8 eV obtained were assigned to the C-C (nonoxygenated carbon) and C-O (oxidized carbon) species, respectively. Figure 5 displayed the Mn 2p and O1s spectra for different samples, respectively. The related results of surface element compositions in molar ratio were listed in Table 1. The XPS spectra of Mn 2p presented two contributions, assignable to spin-orbit splitting into Mn $2p_{3/2}$ (641.6 eV) and Mn $2p_{1/2}$ (653.2 eV), respectively. The BE separation between the two main peaks was 11.6 eV [57]. The Mn 2p spectrum of each catalyst could be further resolved into four peak components, attributable to the presence of surface $Mn^{3+}$ species at BE = 641.2 and 652.9 eV and $Mn^{4+}$ species at BE = 642.6 and 654.0 eV [58]. According to Table 1, the surface $Mn^{4+}/Mn^{3+}$ molar ratio of $\alpha$-$Mn_2O_3$-C was significantly higher than the other two catalysts, indicating that $\alpha$-$Mn_2O_3$-C with (001) crystal facets possessed more surface $Mn^{4+}$ species. The differences in surface $Mn^{4+}/Mn^{3+}$ molar ratios of three catalysts may be explained by the difference in morphologies and the degree of coordination unsaturation of surface active atoms on exposed crystal facets.

As revealed in Figure 5B, the O1s spectra of different $\alpha$-$Mn_2O_3$ catalysts were decomposed into two components at BE = 529.6 and 531.2 eV, corresponding to the surface lattice oxygen ($O_{latt}$: $O^{2-}$) and $O_{ads}$ such as $O^-$, $O_2^-$, $O_2^{2-}$ and $OH^-$ [37,58–60], respectively. The surface $O_{ads}/O_{latt}$ molar ratios of $\alpha$-$Mn_2O_3$-C, $\alpha$-$Mn_2O_3$-TO, and $\alpha$-$Mn_2O_3$-O were 0.53, 0.43, and 0.38, respectively (Table 1). Obviously, the amount of surface adsorbed oxygen in $\alpha$-$Mn_2O_3$-C with (001) facets is much higher than that of $\alpha$-$Mn_2O_3$-TO with (111) & (001) and $\alpha$-$Mn_2O_3$-O with (111) facets, implying the important role of exposed (001) crystal facet in oxygen vacancy formation. The increase in surface adsorbed oxygen species might have a significant effect in enhancing catalytic performance for total oxidation reactions mainly because the surface oxygen species had higher mobility than lattice oxygen in the presence of abundant oxygen vacancies [61–63].

Oxygen vacancy concentration is crucial in combustion reaction, and oxygen can be chemisorbed on the site of oxygen vacancy to become adsorbed reactive species under reaction atomosphere. Russo et al. [59] have directly detected the existence of oxygen vacancy by Raman characterization, agreeing with the XPS characterization results of surface adsorbed oxygen species.

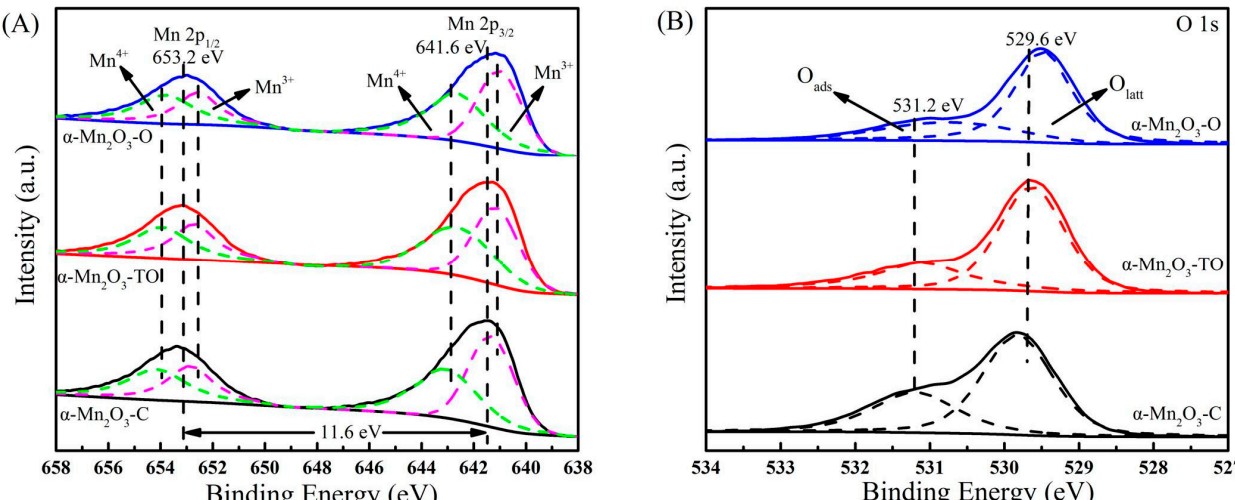

**Figure 5.** XPS patterns of the Mn 2p (**A**) and O 1s (**B**) of $\alpha$-$Mn_2O_3$ catalysts with different morphologies.

In order to further unravel the origin of the morphology effect of $\alpha$-$Mn_2O_3$, CO-TPSR was also performed by using CO as probe molecule over online MS. The curves of the $CO_2$ ($m/z = 44$) signals for three catalysts were displayed in Figure 6. The positions of $CO_2$ peaks can be related to the different reactivity of the surface reactive oxygen species to oxidize CO. Figure 6 clearly showed the different capability for $CO_2$ generation by surface active oxygen species over the samples, which was ranked as the decreased order of $\alpha$-$Mn_2O_3$-C with (001) facets > $\alpha$-$Mn_2O_3$-TO with (001) & (111) facets > $\alpha$-$Mn_2O_3$-O with (111) facets. It was evident that the $\alpha$-$Mn_2O_3$-C sample with exposed (001) could afford the much more abundant surface reactive oxygen species for $CO_2$ generation than the other two samples with exposed (111) and mixed (001) & (111) facets, agreeing well with the sequence of the $H_2$-TPR and XPS results discussed above. The results demonstrated the $\alpha$-$Mn_2O_3$-C with (001) facets contains more lattice oxygen and surface oxygen species as activated oxygen species for CO oxidation than (111) facets.

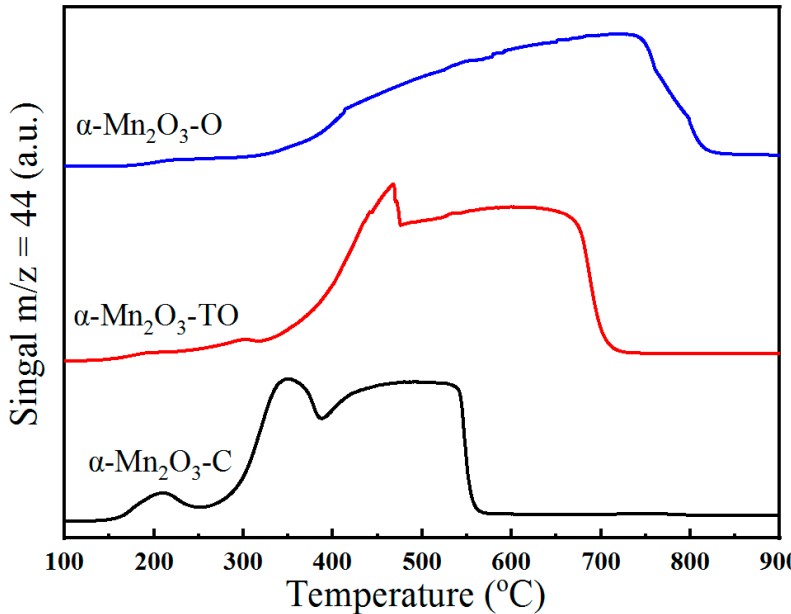

**Figure 6.** Morphology-dependent CO-TPSR profiles of $\alpha$-$Mn_2O_3$ catalysts.

$H_2$-TPR and CO-TPSR have provided a great deal of practical information on the reactivity of surface oxygen species towards hydrogen molecule. However, the information is indirect as to how strongly the surface and lattice oxygen held in $\alpha$-$Mn_2O_3$ catalysts

reacts with substrate molecule i.e., ethanol. Ethanol-TPD is a much more direct measure for the determination of surface oxygen species as compared to $H_2$-TPR. Hence, the catalytic nature of surface adsorbed reactive oxygen species and the desorption behavior of ethanol on the morphology-controlled $\alpha$-$Mn_2O_3$ catalysts with different exposed crystallographic facets has been investigated by using ethanol-TPD technique. Figure 7 showed the ethanol-TPD profiles of different morphologies $\alpha$-$Mn_2O_3$ catalysts. As the desorption temperature rises, the ethanol reactant as well as some important intermediates including ethanol, acetaldehyde, $CO_2$, and $H_2O$ were detected on mass spectrometer, corresponding to the signals of $m/z = 31$, $m/z = 29$, $m/z = 44$, and $m/z = 18$, respectively. It was noticed that the dehydrogenation of ethanol occurred already over three samples in the low temperature range of 40–200 °C, as evidenced by the characteristic peak of acetaldehyde at $m/z = 29$. TPD profiles for $\alpha$-$Mn_2O_3$ catalysts show an apparent morphology-dependent $CO_2$ desorption profile at temperature above 160 °C. The significantly higher intensity as well as the lowered desorption temperature of $CO_2$ over $\alpha$-$Mn_2O_3$-C catalyst strongly suggest that the preferential exposure of (001) crystal facet by tailoring the catalyst morphology may favor the generation of more abundant surface oxygen reactive species and therefore facilitate $CO_2$ production.

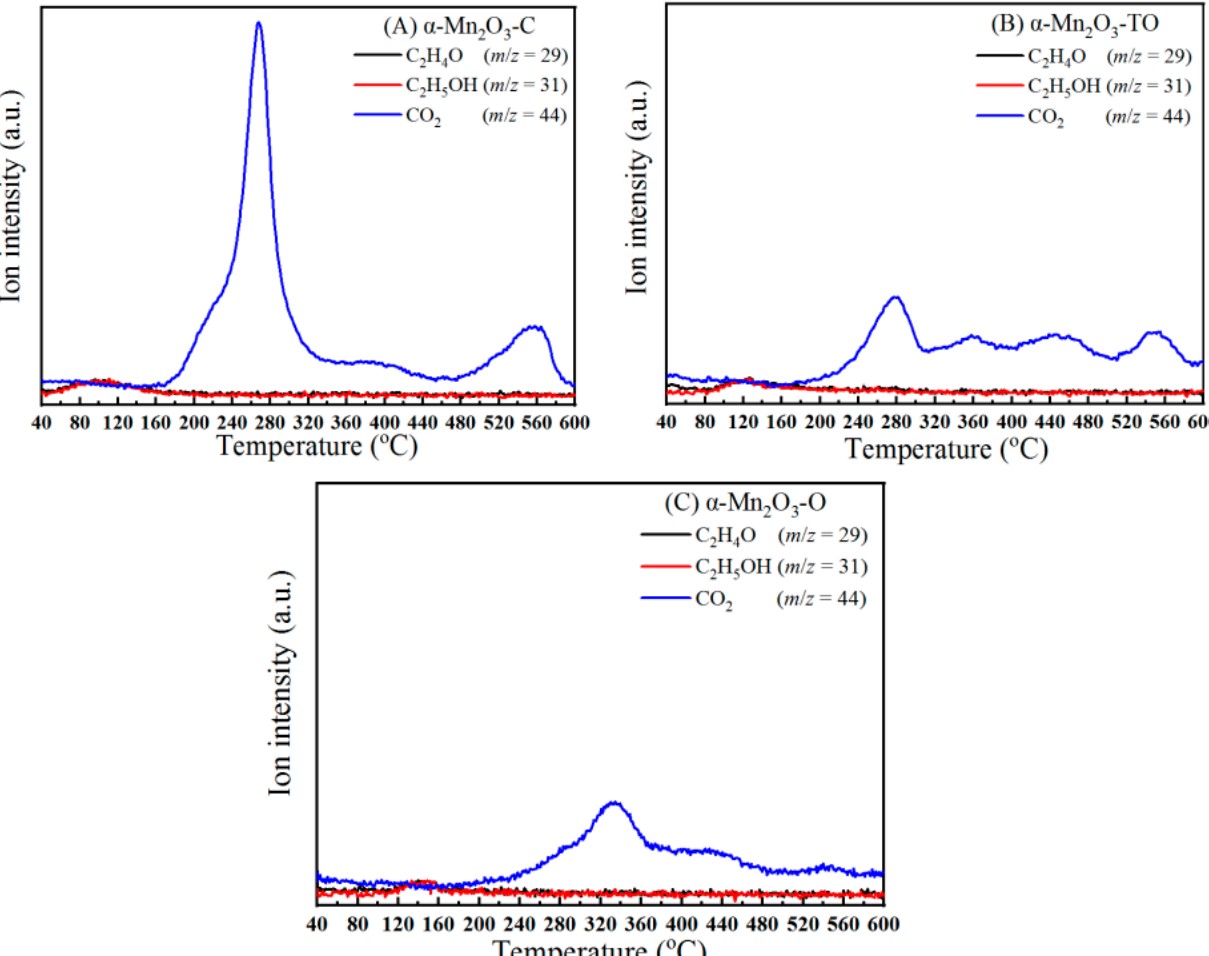

**Figure 7.** Morphology-dependent ethanol-TPD profiles of $\alpha$-$Mn_2O_3$ catalysts: $\alpha$-$Mn_2O_3$-C (**A**), $\alpha$-$Mn_2O_3$-TO (**B**), $\alpha$-$Mn_2O_3$-O (**C**).

### 2.3. Catalytic Performance of Different Shaped $\alpha$-$Mn_2O_3$ Catalysts

The morphology-dependent catalytic performances of $\alpha$-$Mn_2O_3$-C, $\alpha$-$Mn_2O_3$-TO, and $\alpha$-$Mn_2O_3$-O catalysts were evaluated for ethanol total combustion and the results were shown in Figure 8, Figure S4 and Table 2. Figure 8A shows a clear morphology-dependent

catalytic performance for ethanol total combustion with the following order: $\alpha$-$Mn_2O_3$-C (190 °C) > $\alpha$-$Mn_2O_3$-TO (290 °C) > $\alpha$-$Mn_2O_3$-O (340 °C). $\alpha$-$Mn_2O_3$-C catalyst with exposed (001) facets exhibited the best catalytic performance on ethanol total oxidation, achieving the complete conversion temperature at 190 °C, which was 150 °C lower than $\alpha$-$Mn_2O_3$-O catalyst with exposed crystal (111) facets. The same order of $CO_2$ yield was found in Figure 8B, exhibiting nearly 100% $CO_2$ yield at temperature of 240 °C over $\alpha$-$Mn_2O_3$-C, 350 °C over $\alpha$-$Mn_2O_3$-TO, and 400 °C over $\alpha$-$Mn_2O_3$-O catalysts, respectively. Figure 8C showed the volcano-like plot of the acetaldehyde yield over three catalysts, presenting the maximum in acetaldehyde yield of 35.4% at 160 °C over $\alpha$-$Mn_2O_3$-C, 45.8% at 230 °C over $\alpha$-$Mn_2O_3$-TO, and 42.2% at 300 °C over $\alpha$-$Mn_2O_3$-O catalyst, respectively, strongly suggesting that acetaldehyde is the primary intermediate species during ethanol total oxidation, in good accordance with previous work [5,25,28]. It was worth noting that acetaldehyde and $CO_2$ were the only detected carbon-containing products. On the basis of product distribution observed, we concluded that the cascade reaction pathway of ethanol total combustion via acetaldehyde as important intermediate over different crystal facets of $\alpha$-$Mn_2O_3$. Figure 8D and Table 2 further listed the important catalytic data of different $\alpha$-$Mn_2O_3$ catalysts with different exposed crystal facets in terms of $T_{10}$, $T_{50}$, and $T_{90}$, which were defined as the values of the reaction temperature corresponding to the ethanol conversions of 10%, 50%, and 90%, respectively. In this study, $T_{10}$ for $\alpha$-$Mn_2O_3$-C, $\alpha$-$Mn_2O_3$-TO, and $\alpha$-$Mn_2O_3$-O catalysts were 90, 151, and 179 °C; $T_{50}$ were 140, 208, and 261 °C; and $T_{90}$ were 178, 259, and 314 °C, respectively. In addition, the normalized ethanol combustion rate of $\alpha$-$Mn_2O_3$-C, $\alpha$-$Mn_2O_3$-TO, and $\alpha$-$Mn_2O_3$-O at 90 °C were about 2.84, 1.53, and 0.84 mmol·min$^{-1}$·m$^{-2}$·$10^4$, respectively. The superior ethanol combustion activity for $\alpha$-$Mn_2O_3$-C catalyst with exposed (001) crystal facets than $\alpha$-$Mn_2O_3$ with only (111) crystal facets can be closely related to the low-temperature reducibility and highly active surface mobile oxygen species over the $\alpha$-$Mn_2O_3$ surface.

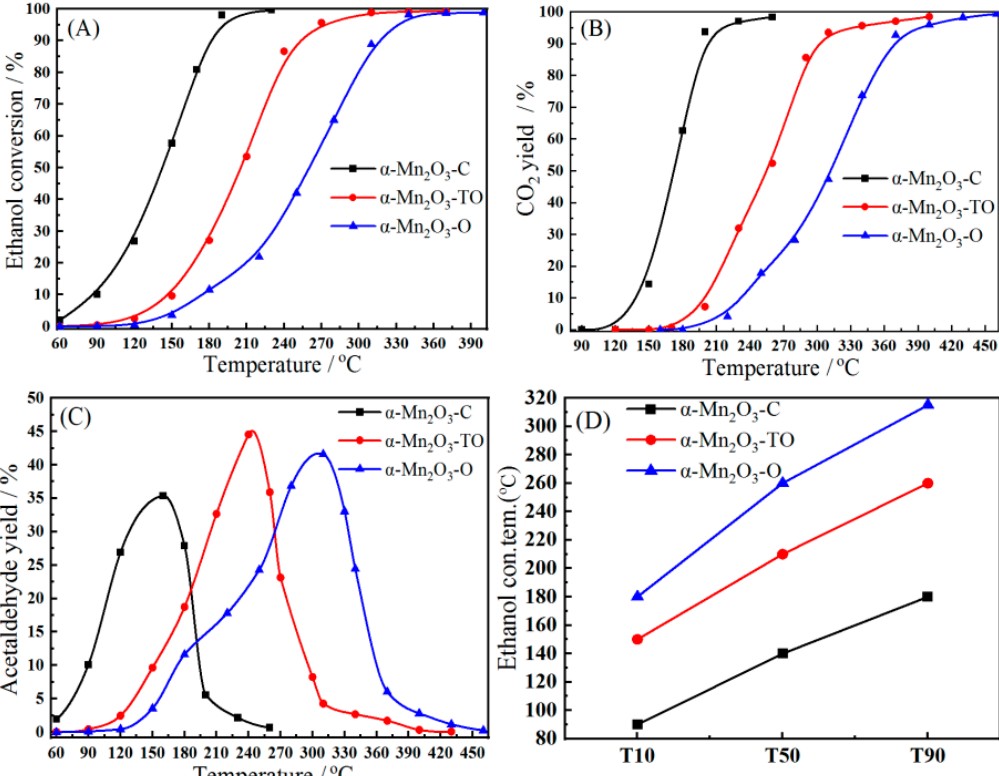

**Figure 8.** Ethanol conversion (**A**), $CO_2$ yield (**B**), and acetaldehyde yield (**C**) of ethanol catalytic performance over $\alpha$-$Mn_2O_3$ catalysts with different morphologies. (**D**) T10, T50, and T90: defined as the temperature approaching 10%, 50%, and 90%, ethanol conversion with the same space velocity over the three catalysts.

**Table 2.** The catalytic activity data of the $\alpha$-Mn$_2$O$_3$ catalysts with different exposed crystal facets for ethanol oxidation under the same reaction conditions: 600 ppm ethanol, 20 vol.% O$_2$, N$_2$ as balance gas, SV = 192,000 mL/(g·h).

| Catalysts | Catalytic Activity (°C) | | | T (°C) | Ethanol Conversion (%) | Normalized Rate (mmol·min$^{-1}$·m$^{-2}$) × 10$^4$ | E$_a$ (kJ/mol) | R$^2$ for E$_a$ |
|---|---|---|---|---|---|---|---|---|
| | T$_{10}$ | T$_{50}$ | T$_{90}$ | | | | | |
| $\alpha$-Mn$_2$O$_3$-C | 90 | 140 | 178 | 90 | 10.1 | 2.84 | 55.3 | 0.99 |
| $\alpha$-Mn$_2$O$_3$-TO | 151 | 208 | 259 | 90 | 0.50 | 1.53 | 63.7 | 0.99 |
| $\alpha$-Mn$_2$O$_3$-O | 179 | 261 | 314 | 90 | 0.10 | 0.84 | 68.3 | 0.98 |

The BET surface area is an important parameter in determining the catalytic performance of heterogeneous catalysts [64–66]. Herein, to eliminate the effect of the BET surface areas, the kinetics of ethanol catalytic oxidation were investigated. And Arrhenius plots were obtained over $\alpha$-Mn$_2$O$_3$ catalysts with different exposed crystal facets in the kinetically controlled regime. The apparent activation energy (E$_a$) was calculated via Arrhenius equation by normalizing reaction rate on per unit BET surface area. The results in Figure 9A and Table 2, showed that $\alpha$-Mn$_2$O$_3$-C catalyst exhibited the much higher initial normalized ethanol reaction rate at low temperature compared with $\alpha$-Mn$_2$O$_3$-TO, and $\alpha$-Mn$_2$O$_3$-O catalysts, in the reverse order of the apparent reaction activation energy. The lower the E$_a$ values indicate the easier oxidation of ethanol, and hence the better performance of a catalyst [1,31,32,34,67]. Under the same reaction conditions, (Table 2) the E$_a$ values of $\alpha$-Mn$_2$O$_3$-C, $\alpha$-Mn$_2$O$_3$-TO, and $\alpha$-Mn$_2$O$_3$-O catalysts were 55.3 kJ/mol, 66.7 kJ/mol, and 68.3 kJ/mol, respectively. The observed morphology effect of $\alpha$-Mn$_2$O$_3$ on the catalytic activity is presumably related to the nature of exposed crystal facets, and density of surface Mn$^{4+}$ and surface reactive oxygen species, which are responsible for the adsorption/activation of ethanol and oxygen molecules. Hence, $\alpha$-Mn$_2$O$_3$-C exhibited superior catalytic performance than $\alpha$-Mn$_2$O$_3$-TO and $\alpha$-Mn$_2$O$_3$-O catalysts, and further illustrated cubic with (001) facets was more active followed by octahedra with (111) facets, in consistent with our XPS, CO-TPSR, and ethanol-TPD results discussed earlier.

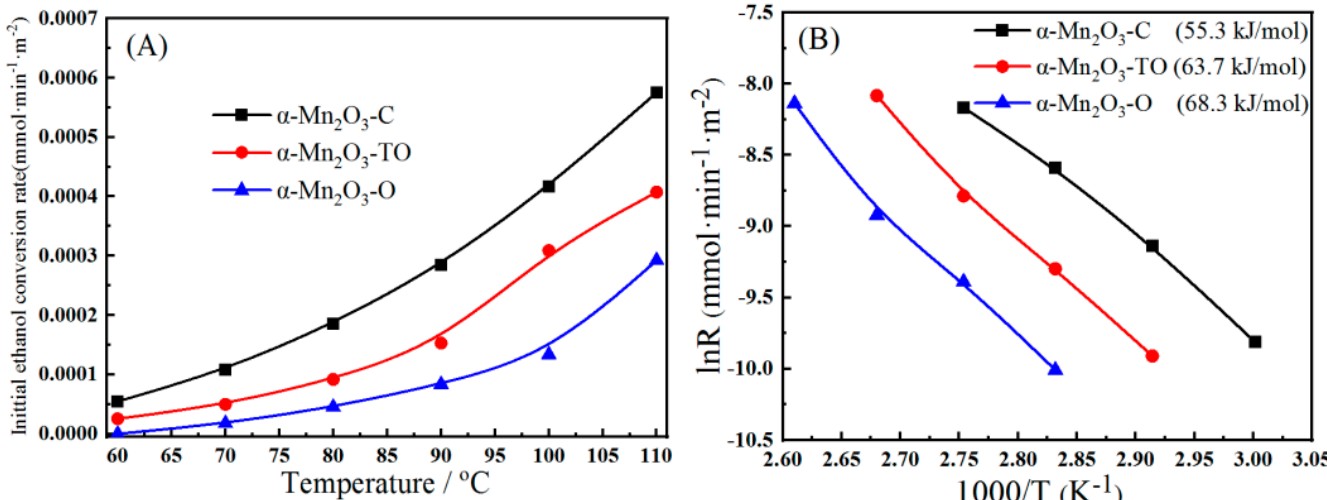

**Figure 9.** Initial ethanol conversion rates (**A**) and Arrhenius plots in ethanol oxidation (**B**) of $\alpha$-Mn$_2$O$_3$ catalysts with different morphologies.

As documented in the literature, the dissociation of oxygen molecules on can be seen as the first elementary step of catalytic oxidation over metal or oxide surfaces. Therefore, the activation capability of oxygen molecules play an important role in surfaces of oxidation catalyst. For metal oxide catalyst, oxygen molecules can be facilely adsorbed and activated in the presence of oxygen vacancies, forming surface adsorbed reactive oxygen species, which would participate in the oxidation of organic substrates. Recently, a combined experimental/theoretical study has been performed by our group [32] to investigate the

morphology–dependent reactivity of $\alpha$-$Mn_2O_3$ on soot combustion and gained the origin of the difference on their crystal facet-dependent reactivity. The density functional theory (DFT) calculations revealed that the oxygen atoms on top-layer of crystal (001) facets on $\alpha$-$Mn_2O_3$ catalysts can flexibly move with the oxygen atoms of sub-layer. By contrary, such a transfer is energetically unfavorable over the crystal (111) facets of $\alpha$-$Mn_2O_3$. Moreover, the formation energy of oxygen vacancy over (001) crystal facet and the coordination number (CN) of surface oxygen atoms is remarkably lower than that on (111) surface. As pointed by An et al. [68], the enhanced coordination unsaturation on specially exposed crystal facet may facilitate oxidation reactions. Accordingly, abundant surface $Mn^{4+}$ ions and surface adsorbed reactive oxygen species over (001) surface of $\alpha$-$Mn_2O_3$ catalysts may facilitate ethanol/oxygen activation at low temperature and boost ethanol combustion efficiency, in good accordance with our activity and characterization results.

### 2.4. Effects of SV, Ethanol, and Water Vapor Concentration

The influence of SV, ethanol, and water vapor concentration was investigated over the so-far best-performing $\alpha$-$Mn_2O_3$-C catalysts and presented in Figure 10. The effect of SV in Figure 10A showed that the catalytic performance increased as the SV decreased, and the temperatures for complete ethanol oxidation to $CO_2$ also shifted from 230 °C down to 180 °C. The temperature values corresponding to the maximum acetaldehyde yield also decreased from 160 °C to 150 °C, indicating that the activity of the sample increased as the space velocity decreased. The influence of the initial concentration of ethanol was studied and depicted in Figure 10B. Under the reaction conditions of initial ethanol concentration = 1200 ppm, $\alpha$-$Mn_2O_3$-C catalyst showed the catalytic activity of $T_{10}$ = 99 °C, $T_{50}$ = 156 °C, and $T_{90}$= 203 °C which was inferior to those ($T_{10}$ = 90 °C, $T_{50}$ = 140 °C, and $T_{90}$ = 178 °C) obtained from the initial ethanol concentration = 600 ppm as shown in Table 3 and Figure 10B. The impact of steam was also investigated over $\alpha$-$Mn_2O_3$-C catalyst. Figure 10C clearly showed that the oxidation rate of ethanol could be considerably suppressed by the presence of steam in the feed, in accordance with the inhibition effect of steam on VOCs combustion in the literatures [28,30,69]. This is not unexpected because the strong and competitive adsorption of water vapor on the catalyst surface may lead to a decrease in the available active sites towards VOCs and oxygen, and therefore inhibit ethanol combustion reaction. Under the reaction conditions of 6 vol.% $H_2O$ in the feed, the catalytic activity ($T_{10}$ = 150 °C, $T_{50}$ = 195 °C, and $T_{90}$ = 218 °C) over $\alpha$-$Mn_2O_3$-C catalyst became much worse than that under conditions without $H_2O$ ($T_{10}$ = 90 °C, $T_{50}$ = 140 °C, and $T_{90}$ = 178 °C), as shown in Table 3 and Figure 10D.

### 2.5. The Stability of $\alpha$-$Mn_2O_3$-C Catalyst

Figure 11 displayed the ethanol catalytic combustion performance as a function of time on stream over the best-performing $\alpha$-$Mn_2O_3$-C catalyst with exposed (001) crystal facets under the conditions of reaction temperature 230 °C, SV 192,000 mL/(g·h), and feed composition of 600 ppm ethanol, 20 vol.% $O_2$, 6 vol.% $H_2O$, $N_2$ as balance gas. Apparently, with the addition of 6 vol.% $H_2O$, the $\alpha$-$Mn_2O_3$-C catalyst exhibited outstanding reaction stability for ethanol oxidation was assessed with experiments lasting at over 230 °C. According to the stability data of successive measurements (Figure 11A), the ethanol conversion, acetaldehyde yield, and $CO_2$ yield remained almost unchanged for the test duration of more than 50 h, illustrating the excellent stability over the entire run of the stability test. Figure 11B presented almost the same XRD diffraction peaks on the used $\alpha$-$Mn_2O_3$-C catalyst as the pattern on the fresh example. Furthermore, the FE-SEM inset in Figure 11B illustrated that the $\alpha$-$Mn_2O_3$-C sample remained their original cubic morphology after the test duration of more than 50 h. Based on the stability test, it appeared that $\alpha$-$Mn_2O_3$-C could be used as one of the promising manganese based catalysts for efficient ethanol combustion under practical conditions.

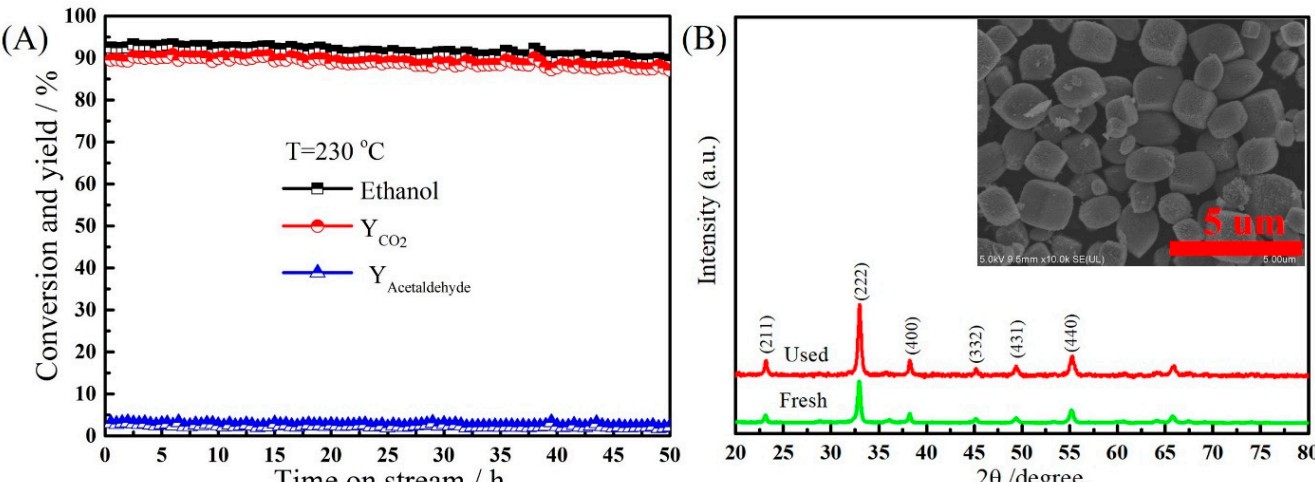

**Figure 10.** The catalytic performances of the α-Mn₂O₃-C catalyst under different reaction conditions of ethanol combustion, the effect of performance for (**A**) space velocity: 96,000 mL/(g·h) and 192,000 mL/(g·h). (**B**) ethanol concentration: 600 ppm and 1200 ppm. (**C**) steam: 6 vol.% H₂O. (**D**) T10, T50, and T90: the temperature over the α-Mn₂O₃-C catalyst with different conditions where ethanol conversion approaches 10%, 50%, and 90%, respectively.

**Figure 11.** (**A**) The stability test of the α-Mn₂O₃-C catalyst for ethanol total combustion. (**B**) XRD patterns of used and fresh α-Mn₂O₃-C catalysts, FE-SEM image of used α-Mn₂O₃-C catalyst. Reaction conditions: reaction temperature 230 °C, 6 vol.% H₂O, ethanol concentration 600 ppm, 20 vol.% O₂, N₂ balance, SV = 192,000 mL/(g·h).

**Table 3.** The catalytic performances of the $\alpha$-$Mn_2O_3$-C catalyst under different reaction conditions of ethanol oxidation.

| Ethanol Con. Tem (°C) | 600 ppm Ethanol SV of 96,000 mL/(g·h) | 600 ppm Ethanol SV of 192,000 mL/(g·h) | 1200 ppm Ethanol SV of 192,000 mL/(g·h) | 600 ppm Ethanol, 6 vol.% $H_2O$, SV of 192,000 mL/(g·h) |
|---|---|---|---|---|
| $T_{10}$ | 79 | 90 | 99 | 150 |
| $T_{50}$ | 136 | 140 | 156 | 195 |
| $T_{90}$ | 172 | 178 | 203 | 218 |

## 3. Materials and Methods

### 3.1. Catalyst Synthesis

Three types of morphology-tuned $\alpha$-$Mn_2O_3$ catalysts, including morphology of cubic, truncated octahedra and octahedra, were synthesized via a low-temperature hydrothermal method according to the literatures earlier [32,41,44,45]. For truncated octahedra and octahedra $\alpha$-$Mn_2O_3$ catalysts was prepared by solvothermal method with different solvents. The first catalyst $\alpha$-$Mn_2O_3$-octahedra (refer as $\alpha$-$Mn_2O_3$-O), for a typical synthesis, $Mn(NO_3)_2 \cdot 4H_2O$ (16 mmol) was dissolved in ethanol (52 mL) at room temperature (RT) under vigorous magnetic stirring for 20 min to form a homogeneous solution, which then then transferred to a 60 mL Teflon-lined stainless-steel autoclave. The autoclave was tightly sealed and heated for 10 h at 383 K. After hydrothermal reaction, the autoclave was cooled down to RT naturally. The resulting solid product was centrifuged, washed three times in distilled water and ethanol to eliminate impurity ions, and then placed into an oven at 373 K overnight to obtain the $\alpha$-$Mn_2O_3$-O catalyst. Similarly, the synthetic method of the second catalyst, $\alpha$-$Mn_2O_3$-truncated octahedra (referred to as $\alpha$-$Mn_2O_3$-TO), merely changed the solvent replaced by 2-butanol. All catalysts should be calcined in air atmosphere for 2 h at 873 K. The third catalyst, $\alpha$-$Mn_2O_3$-cubic (refer as $\alpha$-$Mn_2O_3$-C), glucose (6 mmol) was added into a $KMnO_4$ solution (6 mmol of $KMnO_4$ was dissolved in 60 mL of distilled water) at RT under vigorous magnetic stirring for another 20 min to form a homogeneous solution, which then was transferred to a 100 mL autoclave, which was tightly sealed and heated for 10 h at 433 K, after natural cooling to RT, immediately followed via centrifugation, washing, and final drying to obtain the precursor $MnCO_3$. Prior to yielding cubic $\alpha$-$Mn_2O_3$ catalyst, the precursor $MnCO_3$ should be calcined in air atmosphere for 2 h at 873 K with a ramp rate of 5 K/min. Finally, a series of $\alpha$-$Mn_2O_3$ catalysts were crushed and sieved to 40–60 mesh for catalytic activity tests.

### 3.2. Catalyst Characterization

The crystalline structure of catalysts was recorded by the powder X-ray diffraction (XRD) on a Bruker D2 Phaser (Bruker Axs Gmbh, Karlsruhe, Germany) using Cu-K$\alpha$ radiation. The $2\theta$ of the XRD range from 15º to 80° with the scan speed during analysis was 0.5 s/step. Field emission-scanning electron microscopy (FE-SEM) surface morphology of the catalyst on Hitachi S-4800 instrument (Hitachi, Ibaraki, Japan). High-resolution transmission electron microscopy (HR-TEM) images of the catalysts were taken on Hitachi JMF-2100 instrument (Hitachi, Japan). The Fourier transform infrared (FT-IR) spectrum of the catalyst was collected by using a Nicolet 380 spectrometer (Nicolet, Madison, WI, USA) in the range of 400–4000 $cm^{-1}$. The nitrogen adsorption−desorption isotherms were performed on a Micromeritics ASAP 2460 (Micromeritics, Norcross, GA, USA) analyzer at 77 K. By using Brunauer-Emmet-Teller (BET) and Barrett-Joyner-Halenda (BJH) models, specific surface areas ($S_{BET}$), pore diameter ($D_P$) and pore volume ($V_P$) of the catalysts were calculated from the adsorption branches of the isotherms. Prior to measurement it was degassed at least for 5 h at 393 K. The X-ray photoelectron spectroscopy (XPS) measurements of the catalyst was examined on ESCALAB 250 Xi (Thermo Fisher Scientific, Waltham, MA, USA) using Al-K$\alpha$ X-ray source. The binding energy scale was corrected for surface charging by use of the C 1s peak of contaminant carbon as reference at 284.6 eV.

Hydrogen temperature-programmed reduction ($H_2$-TPR) measurements were carried out on a chemisorption analyzer (Micromeritics, Auto Chem II2920, Norcross, GA, USA)

instrument, which was equipped with a thermal conductivity detector (TCD) to calibrate the $H_2$ consumption of metal valence reduction. Before $H_2$-TPR experiments started, the catalysts (50 mg, 40–60 mesh) were loaded into a U-shaped fixed-bed quartz micro-reactor and pretreated in an Ar atmosphere of 50 mL/min for 1 h at 573 K, and then cooled down to 373 K. The pretreated catalysts were reduced under a mixture-gas of 10 vol.% $H_2$–90 vol.% Ar flow with 30 mL/min and heated in the range of 373 K to 1173 K with a ramp rate of 10 k/min. Ethanol temperature-programmed desorption (Ethanol-TPD) measurements were carried out via using a Micromeritics AutoChem II2920 instrument connected to mass spectrometry (MS) (Hiden, HPR20, Warrington, UK) instrument. The catalysts (50 mg, 40–60 mesh) were loaded into a U-shaped fixed-bed quartz micro-reactor and previously treated for 0.5 h at 373 K in pure He flow with 30 mL/min, cooling down to 313 K. The adsorption experiment of ethanol was carried out with a He flow of 30 mL/min. After saturation, the temperature was increased from 313 K to 873 K with a ramping rate of 10 K/min. The desorption substances coming from $\alpha$-$Mn_2O_3$ catalysts were monitored by means of online MS apparatus.

### 3.3. Temperature-Programmed Surface Reactions (TPSR) and Mixed-Gas without Oxygen

The CO-TPSR measurements(Micromeritics, Auto Chem II2920, America ) to characterize the nature of the active sites of catalysts on $\alpha$-$Mn_2O_3$ catalysts were performed in a fixed-bed tubular quartz system ($\Phi$ = 6.0 mm) via using MS, where masses (m/e: CO = 28, $CO_2$ = 44) were monitored. 50 mg of each catalyst were previously treated for 0.5 h at 773 K in a pure Ar flow of 50 mL/min, in order to remove physical adsorption oxygen, cooling down to 373 K. The mixture-gas of 5 vol.% CO-95 vol.% He with a total flow rate of 30 mL/min increasing the temperature from 373 K to 1173 K, and a ramping rate of 5 K/min.

### 3.4. Catalytic Performance Evaluation

The morphology-dependent catalytic performance of three $\alpha$-$Mn_2O_3$ catalysts for ethanol total combustion were evaluated in a flow-through quartz micro-reactor ($\Phi$ = 6.0 mm), which was positioned in the center of the tube furnace at atmospheric pressure. The catalytic combustion temperature was monitored via a thermocouple, which was placed in a few millimeters above the bottom of the catalyst fixed-bed. 100 mg (40–60 mesh) of catalyst was held in place by using quartz wool at upper and lower ends [70,71]. The total flow rate of the uniformity mixed-gas (600 ppm ethanol/20 vol.% $O_2$/$N_2$ balance gas) passed through the catalyst fixed-bed remained at 320 mL/min, which was detected by using mass flow-meter. The space velocity (SV) of catalytic ethanol combustion was 192,000 mL/(g × h). Prior to catalytic evaluation, to exclude the effect of impurities, the catalysts should be pretreated for 30 min at 333 K in air atmosphere. The outflow gases of the oxidation reaction from the reactor were quantitatively analyzed online via a Gas Chromatograph (Shimadzu, GC-2014C, Kyoto, Japan) which was equipped with two detectors of a thermal conductivity detector (TCD) and a flame ionization detector (FID). And TCD was only responsible for the detection of $CO_2$, FID responsible for the detection of ethanol and acetaldehyde due to it can respond to almost all organic matter. In the present work, the final products of ethanol total combustion were $CO_2$ and $H_2O$. During the ethanol oxidation, the main products were acetaldehyde, $CO_2$ and other remaining trace products which can be ignored. The effect of performance for SV, ethanol and water vapor concentration were explored. The thermal stability of $\alpha$-$Mn_2O_3$ catalyst was tested for 50 h under the condition of 6.5 mg catalyst was diluted with 100 mg of $SiO_2$, 600 ppm ethanol, 6vol.% $H_2O$, 20 vol.% $O_2$, $N_2$ as the equilibrium gas, and SV was 192,000 mL/(g × h). The conversion of ethanol was calculated based on the participation of ethanol as follows (1):

$$X = \left( \left( C_{\text{acetaldehyde out}} + \frac{C_{co_2 \text{ out}}}{2} \right) \Big/ \left( C_{\text{acetaldehyde out}} + \frac{C_{co_2 \text{ out}}}{2} + C_{\text{ethanol out}} \right) \right) \times 100\%$$

$$S_{co_2} = \left( \frac{C_{co_2 \text{ out}}}{2} \Big/ \left( C_{\text{acetaldehyde out}} + \frac{C_{co_2 \text{ out}}}{2} + C_{\text{ethanol out}} \right) \right) \times 100\%$$

$$S_{\text{acetaldehyde}} = C_{\text{acetaldehyde out}} \Big/ \left( C_{\text{acetaldehyde out}} + \left( \frac{C_{co_2 \text{ out}}}{2} \right) \right) \times 100\%$$

$$Y_{\text{yield}} = \text{conversion} \times \text{selectivity} \times 100\%, \tag{1}$$

where X, ethanol conversion rate; C, concentration; S, selectivity of products; Y, yield of products.

### 3.5. Kinetic Analysis for Ethanol Total Oxidation

To obtain kinetics data, the kinetic research was performed under ethanol conversion below 15% with the condition of 600 ppm ethanol, 20 vol.%$O_2$, $N_2$ as the equilibrium gas, and SV was 192,000 mL/(g·h) for three $\alpha$-$Mn_2O_3$ catalysts in the range of 333 K to 383 K. The catalytic activation energy ($E_a$) of the catalysts were calculated according to the Arrhenius Equation (2):

$$\ln R = -A \exp\left(-\frac{E_a}{RT}\right), \tag{2}$$

where $E_a$, apparent activation energy (kJ/mol); R, the reaction rate (mol·min$^{-1}$·m$^{-2}$); T, the value of the reaction temperature (K).

### 4. Conclusions

In summary, a series of $\alpha$-$Mn_2O_3$ catalysts with different morphologies, including cubic, truncated octahedra and octahedra, were controllably synthesized by using a facile hydrothermal process and systematically investigated for their morphology-dependent catalytic performance in ethanol total combustion. The catalytic activity according to each manganese oxide catalyst was in the order of $\alpha$-$Mn_2O_3$-C > $\alpha$-$Mn_2O_3$-TO > $\alpha$-$Mn_2O_3$-O. The present study indicated that this morphology-dependent reactivity of $\alpha$-$Mn_2O_3$ nanocrystal was originated from the chemical nature of the exposed (001) facets. As revealed by characterization results of HR-TEM, $H_2$-TPR, XPS, CO-TPSR, ethanol-TPD, the superior activity of $\alpha$-$Mn_2O_3$-C sample can be well correlated with its enhanced low temperature reducibility, abundant surface $Mn^{4+}$ species and surface reactive oxygen species governed by the specially exposed (001) facets. Moerover, the effect of space velocity and feed compsotion (ethanol and steam) were also investigated on $\alpha$-$Mn_2O_3$-C catalyst under different reaction conditions. Furthermore, the $\alpha$-$Mn_2O_3$-C catalyst exhibited good stability for 50 h at 230 °C in the presence of 6 vol.% $H_2O$, demonstrating that $\alpha$-$Mn_2O_3$-C catalyst could be used as promising catalyst for effcent practical total oxidation of ethanol. This study presents a new strategy to design and develop the catalyst for efficient combustion of ethanol by morphological control of earth-abundant inexpensive Mn-based catalysts.

**Supplementary Materials:** The following supporting information can be downloaded at: https://www.mdpi.com/article/10.3390/catal13050865/s1, Figure S1: FT-IR spectra of $\alpha$-$Mn_2O_3$ catalysts with different morphologies; Figure S2: $N_2$ adsorption-desorption isotherms (A), and the pore size distributions (B) of three $\alpha$-$Mn_2O_3$ catalysts; Figure S3: XPS patterns of the survey (A) and C 1s (B) of different morphologies $\alpha$-$Mn_2O_3$ catalysts; Figure S4: Ethanol conversion, acetaldehyde and $CO_2$ selectivity profiles of ethanol catalytic performance over $\alpha$-$Mn_2O_3$ catalysts with different morphologies. $\alpha$-$Mn_2O_3$-C (A), $\alpha$-$Mn_2O_3$-TO (B), $\alpha$-$Mn_2O_3$-O (C) under normal conditions.

**Author Contributions:** Conceptualization, Y.M. and W.A.; methodology, Y.M.; software, W.L.; validation, W.L., F.J. and Y.M.; formal analysis, W.L., F.J. and J.W.; investigation, W.L.; resources, Y.M.; data curation, W.L., F.S. and S.L.; writing—original draft preparation, W.L., Y.M., W.A. and T.T.M.; writing—review and editing, W.L., W.A., T.T.M. and Y.M.; supervision, Y.M. and T.T.M.; project administration, Y.M.; funding acquisition, Y.M. All authors have read and agreed to the published version of the manuscript.

**Funding:** This research was supported by National Natural Science Foundation of China (22179081, 22076117), Class III Peak Discipline of Shanghai—Materials Science and Engineering (High-Energy Beam Intelligent Processing and Green Manufacturing), and Science and Technology Commission of Shanghai Municipality (20ZR1422500).

**Data Availability Statement:** The data presented in this study are available on request from the corresponding author.

**Acknowledgments:** The authors are grateful to National Natural Science Foundation of China (22179081, 22076117), Class III Peak Discipline of Shanghai—Materials Science and Engineering (High-Energy Beam Intelligent Processing and Green Manufacturing) and Science and Technology Commission of Shanghai Municipality (20ZR1422500) for their support in catalyst research.

**Conflicts of Interest:** The authors declare no conflict of interest.

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
