# Peer review of "Boosting Catalytic Combustion of Ethanol by Tuning Morphologies and Exposed Crystal Facets of α-Mn2O3"

_catalysts, doi:10.3390/catal13050865_

Round 1

Reviewer 1 Report

In this work, Liu et al. prepared several α-Mn2O3-based catalysts with the same cubic phase but with different exposed crystal surfaces. The catalysts were characterized and employed for the combustion of ethanol. The authors attempted to correlate the observed catalytic behavior to the different exposed crystal planes. Overall, whereas the work is potentially interesting I cannot suggest it for inclusion in Catalysts do to concerns about novelty and content. An amended version of the manuscript may be re-evaluated at a later point. 

a) As also the authors admitted in the manuscript, many aspects of this work are very similar to their previous study in reference 39. Figures 1-4 are very very very similar, if not identical, to the corresponding figures in ref. 39. I understand that these are the same materials. But if the authors use the same materials for the same type of reaction (combustion/oxidation) to reach to the same conclusion on the higher activity of the {001} planes, where is the novelty? Just the substrate? 

b) The authors cannot really compare the materials. One has a surface area that is one order of magnitude higher than the others; there is a difference of 30 times between the most and the least active material in terms of surface area. It is true that the number of vacancies, the energy of vacancy formation, the ability to adsorb activated oxygen species, play a role. However, these factors can be compared only on similar surfaces areas. It has already been showed that the surface area is a crucial factor in (total) oxidation reactions (Catal. Today 2020347, 23–30; Catalysts 202212, 1533; Ceram. Int. 201945, 2779–2788). I also see that the authors have shown the different energy of activation values, however, these are "apparent" energies of activation that depends on the initial rates of the experimental reactions and which, in turn, depend on the catalyst surface area. What would it happen if the authors report "surface normalized" initial reaction rates? I guess that the story would totally change. These aspects should be much more systematically considered.

Reviewer 2 Report

The manuscript «Boosting catalytic combustion of ethanol by tuning morphologies and exposed crystal facets of α-Mn2O3» represents study of three types of α-Mn2O3 catalysts with different morphologies (cubic, truncated octahedra and octahedra) and their properties in ethanol total oxidation at low temperatures.  Authors utilized XRD, BET, FE-SEM, HR-TEM, FT-IR, 19 H2-TPR, XPS, ethanol-TPD, and CO-TPSR techniques to characterize manganese oxides. Although, it is an interesting study, there are several aspects that should be reviewed before it can be accepted for publication.

1.       Typos : Line 50, CoFe2O4[20], line 62 , To this end, line 190 α-Mn2O3[57].

2.       Introduction part. «For transition-metal oxides (TMOs), the Langmuir-hinshelwood (LH) and Mars-Van Krevelen (MVK) mechanism are proverbially accepted to be responsible for the oxidationof VOCs; the gas-phase organic molecules is oxidized by the active oxygen species of TMOs, following is re-oxidized by gas-phase oxygen molecules, which will regenerate or maintain the oxidation state of the metal cations[13,14,32]» In this text the Mars-Van Krevelen (MVK) mechanism is described. The place of the Langmuir-hinshelwood (LH) mechanism is not clear. What is the difference between these mechanisms?

3.       From Figure 1 it is not clear if there are additional peaks around 30 degrees , the bar graph overlaps with the markers.

4.       From description of manganese oxide structure, it is not clear the influence of oxide morphology on the XRD pattern.

5.       «From the diffraction profiles of α-Mn2O3-C, α-Mn2O3-TO, and 139 α-Mn2O3-O, one can observe the considerably sharpening of the characteristic diffraction 140 peaks with the increasing relative intensity, indicating the enhanced crystallinity» The meaning of «enhanced crystallinity» is not clear. It may be worth estimating the crystalline size using for example equation Sherrer  for different diffraction peaks

6.       TPR characterization. «The alteration of the reduction behavior might be caused by the presence of crystal (001) facets on truncated octahedra α-Mn2O3 sample.» It is worth explaining how the presence of different faces is related to the absorption of hydrogen and the reduction of manganese oxide.

7.        According to Table 1, Mn2O3 oxides with various morphology have different surface area. Please comment the origin of these phenomena

Round 2

Reviewer 1 Report

Liu et al. have submitted a revised version of their manuscript which I surely found improved but there are still issues to be fixed. I suggest that the manuscript is re-evaluated after major revisions as noted:

a) Now that the content of the manuscript is presented in a clearer way, a few additional issues stand out. In the first place, as acetaldehyde is surely a more valuable product than CO2, I noticed that there is not much attention to acetaldehyde selectivity in the manuscript (For instance in Table 2). The best catalyst is likely not the one that burns EtOH faster, but the one that provides the higher amount of acetaldehyde. As shown in Figure 8C, the catalyst in the red trace seems to represent the best match between T50 value and acetaldehyde selectivity. It is likely that the most active catalyst is also the least selective because it is so active that it favors the combustion of acetaldehyde. Overall, there should be more attention to the selectivity of the reaction. Additionally, in Figure 8c, I cannot understand why the highest points of the curves are not included in the fitting; it looks very strange.

b) I am not so sure about the O1s fitting in the XPS spectrum in Figure 5b. It is fitted by using very broad signals that may not be physically meaningful. In the case of metal oxides, there should surely be an additional signal at about 530.5-531.0 eV that represents the oxygen atoms in a oxygen vacant environment (Journal of Industrial and Engineering Chemistry, 2021, 104, 43-60). Indeed it is the oxygen vacancies to be crucial in combustion processes (Applied Catalysis B: Environmental 2016, 197, 35–46) and not the Oads that is removed when ramping up the temperature. These aspects should be discussed more in detail.

c) As in the revised manuscript the authors claim that "The BET surface area is an important parameter in determining the catalytic performance of heterogeneous catalysts" at Page 12, it would worth to mention relevant references such as Catal. Today 2020, 347, 23–30; Catalysts 2022, 12, 1533; Ceram. Int. 2019, 45, 2779–2788.

The English is ok and clear but there are a few grammatical errors that should be fixed.

Reviewer 2 Report

-

Author Response

Thanks for your comments and suggestions.